# InteMATs: Integrating Granularity-Specific Multilingual Adapters for Cross-Lingual Transfer

**Meizhen Liu[1], Xu Guo[2], Jiakai He[1], Jianye Chen[1], Siu Cheung Hui[2], Fengyu Zhou[1]**

[1]Shandong University, [2]Nanyang Technological University

[1]{meizhen.liu.n,jiakai.he,jyccn}@mail.sdu.edu.cn
[2]{xu008,asschui}@ntu.edu.sg, [1]zhoufengyu@sdu.edu.cn

## Abstract

Multilingual language models (MLLMs) have achieved remarkable success in various cross-lingual transfer tasks. However, they suffer poor performance in zero-shot low-resource languages, particularly when dealing with longer contexts. Existing research mainly relies on full-model fine-tuning on large parallel datasets to enhance the cross-lingual alignment of MLLMs, which is computationally expensive. In this paper, we propose **InteMATs**, a novel approach that integrates multilingual adapters trained on texts of different levels of granularity. To achieve this, we curate a multilingual parallel dataset comprising 42 languages to pre-train sentence-level and document-level adapters under the contrastive learning framework. Extensive experiments demonstrate the effectiveness of InteMATs in improving the cross-lingual transfer performance of MLLMs, especially on low-resource languages. Finally, our comprehensive analyses and ablation studies provide a deep understanding of the high-quality representations derived by InteMATs.

## 1 Introduction

Multilingual language models (MLLMs), such as mBERT (Devlin et al., 2019) and XLMR (Conneau et al., 2019), have achieved remarkable success on a wide range of cross-lingual tasks (Zweigenbaum et al., 2017; Artetxe and Schwenk, 2019a; Nivre et al., 2020; Zhang et al., 2019; Clark et al., 2020; Artetxe et al., 2020a). They are pre-trained on large multilingual data using pretext tasks such as masked language modeling (MLM). To apply MLLMs to downstream tasks, we need to fine-tune a separate copy for each specific task.

However, due to limited data during pre-training, MLLMs may struggle to capture cross-lingual alignment for low-resource languages, resulting in subpar performance. To address this issue, subsequent works (Chi et al., 2021b; Liu et al., 2022;

Chi et al., 2022; Luo et al., 2021; Conneau et al., 2019) propose to enhance MLLM representations by continuing the pre-training on more parallel data using techniques such as translation language modeling (e.g., InfoXLM (Chi et al., 2021a) and XLM-ALIGN (Chi et al., 2021b)), contrastive learning (e.g., mSimCSE (Wang et al., 2022)), and so on (Yang et al., 2020; Feng et al., 2022).

Nevertheless, these approaches often require fine-tuning the entire backbone model, which is computationally expensive and can result in catastrophic forgetting (Kirkpatrick et al., 2017). Moreover, they mainly focus on enhancing sentence-level representations. As a result, their performance gains on document-level cross-lingual tasks are not as impressive as sentence-level ones (Hu et al., 2020), indicating a need for a finer-grained perspective in enhancing cross-lingual alignment.

Motivated by recent advancements in parameter-efficient adaptation for large language models (LLMs) (Ding et al., 2022; Guo and Yu, 2022), we propose a novel approach named **InteMATs**, which stands for Integrating Granularity-specific Multilingual Adapters, to enhance cross-lingual transfer performance of MLLMs. Adapter tuning works by training a set of adapter modules conditioned on a frozen LLM (Houlsby et al., 2019). They can equip MLLMs with new knowledge (Hou et al., 2022) and facilitate language-specific adaptation (Pfeiffer et al., 2020, 2022) without modifying pre-trained parameters. Different from them, we offer a new perspective for enhancing cross-lingual alignment by exploiting a set of multilingual adapters pre-trained on different levels of text granularity.

To obtain these adapters, we curate a multilingual parallel dataset consisting of 42 languages from Wikipedia[1] and process them into the sentence-level corpus ($\mathcal{D}_{\mathrm{ST}}$) and document-level corpus ($\mathcal{D}_{\mathrm{DT}}$). Recent findings (Park et al., 2023) reveal that contrastive learning (CL) tends to ex-

---

[1]https://en.wikipedia.org/wiki/Main_Page

tract global information, whereas masked image modeling (MIM) focuses on capturing local features, which well explained earlier results (Wang et al., 2022). Therefore, we employ CL to train adapters to capture global cross-lingual alignment information to augment the MLLM representations that may initially focus on extracting local information. Specifically, we first train a set of multilingual adapters on $\mathcal{D}_{\text{ST}}$ and $\mathcal{D}_{\text{DT}}$ respectively. Then, we learn to fuse them by incorporating layer-wise fusion modules (e.g., AdapterFusion (Pfeiffer et al., 2021)) and train them on downstream data to determine the fusion weights.

Extensive experiments on five kinds of cross-lingual transfer tasks, including sentence-level and document-level ones, demonstrate that InteMATs significantly improves the cross-lingual transfer performance of MLLMs (i.e., mBERT and XLMR). In particular, InteMATs excels the state-of-the-art baseline by $4\%$ on BUCC (Zweigenbaum et al., 2017), $7\%$ on Tatoeba (Artetxe and Schwenk, 2019a), and $3.5\%$ on TydiQA (Clark et al., 2020). Notably, InteMATs brings a substantial $30\%$ improvement over its backbone MLLMs in low-resource languages that are never seen during pre-training, demonstrating the high quality of the representations learned by InteMATs. We finally conduct a comprehensive analysis on InteMATs to unravel the distribution of contributions on each component and layer-wise impact within InteMATs. We will make our data and model publicly available for future research.

## 2 Related Works

**Cross-lingual Representation Learning** Existing researches mainly adopt the full-model fine-tuning approach with monolingual or parallel corpus to obtain cross-lingual representations. They employ pretext tasks such as masked language modeling (MLM) (Devlin et al., 2019; Chi et al., 2021b), casual language modeling (CLM) (Conneau and Lample, 2019), and translation language modeling (TLM) (Chi et al., 2021a,b), to train MLLMs. However, from the results reported on the XTREME benchmark (Hu et al., 2020), we observe a decrease in cross-lingual transfer performance for MLLMs as the input text length increases (Hu et al., 2020). Previous researches mainly focus on improving cross-lingual alignment for sentence representations (Wang et al., 2022; Feng et al., 2022; Artetxe and Schwenk, 2019b). There is a lack of

research specifically addressing the enhancement of document-level cross-lingual representations.

**Adapters for MLLMs** Recently, Adapter-based approaches (Houlsby et al., 2019; Pfeiffer et al., 2020; Artetxe et al., 2020b; Li and Liang, 2021; Pfeiffer et al., 2022) have gained popularity as a parameter-efficient alternative to traditional fine-tuning for large language models (LLMs). These adapters are inserted between the transformer layers and are learned from data conditioned on a frozen LLM. They learn language-specific transformations to facilitate quick adaptation to new languages (Artetxe et al., 2020b) or new tasks (Pfeiffer et al., 2020). Recent research show that it is possible to inject new knowledge into MLLMs to enhance their cross-lingual representations (Hou et al., 2022). Different from previous research, this paper fills the gap of granularity perspective in cross-lingual alignment via adapter tuning.

**Contrastive Learning for Language Models** Contrastive learning (CL) (Gao et al., 2021; He et al., 2022; Chen et al., 2020a,b) has shown much promise in NLP for its capability of capturing discriminative information in an unsupervised manner. Many research works have adopted this pretext task in their MLLM pre-training, such as mSimCSE (Wang et al., 2022), LASER (Artetxe and Schwenk, 2019b), and InfoXLM (Chi et al., 2021a). However, these approaches require fine-tuning the entire backbone, resulting in non-trivial computational overhead. This paper circumvents this challenge by applying CL for pre-training multilingual adapters while leaving MLLMs frozen.

## 3 Preliminaries

We start by introducing some basic knowledge about adapter tuning and contrastive learning.

**Adapter Tuning:** Adapter tuning (Houlsby et al., 2019) is a parameter-efficient transfer learning technique for adapting large pre-trained models to downstream tasks based on adapters (Rebuffi et al., 2017). These adapters are small, new task-specific modules inserted between the layers of a pre-trained model. Instead of fine-tuning the entire model, adapter tuning only trains the parameters of the adapter modules while keeping the pre-trained model fixed. This approach allows us to specialize a pre-trained model in different tasks while retaining the knowledge acquired during pre-training.

Following Houlsby et al. (2019), we use $\phi_{\boldsymbol{w}}$ to denote a pre-trained model with parameters $\boldsymbol{w}$: $\phi_{\boldsymbol{w}}(\boldsymbol{x})$. For adapter tuning, a new function, $\psi_{\boldsymbol{w},\boldsymbol{v}}(\boldsymbol{x})$, is composed, where parameters $\boldsymbol{v}$ denote all the adapter modules and $\boldsymbol{w}$ is copied from pre-trained weights. The architecture of an adapter module is shown in Figure 1 (a). It consists of two feed-forward layers and a non-linear activation function. For the hidden states $\boldsymbol{h}_l$ at layer $l$, an adapter module works as follows:

$$A_l(\boldsymbol{h}_l) = \boldsymbol{W}_l^1(\text{ReLU}(\boldsymbol{W}_l^2 \boldsymbol{h}_l)) + \boldsymbol{r}_l, \quad (1)$$

where $\boldsymbol{W}_l^2$ represents the down-projection matrix, $\boldsymbol{W}_l^1$ represents the up-projection matrix, and ReLU is the activation function. $\boldsymbol{r}_l$ represents the residual information from the original input, which by-passes the adapter's transformations.

**InfoNCE:** InfoNCE (InfoMax with Noise Contrastive Estimation) (Oord et al., 2018) is a loss function commonly used in self-supervised learning, particularly in contrastive learning methods. The goal is to maximize the mutual information between related samples while minimizing it between unrelated samples, facilitating the discovery of discriminative features in an unsupervised manner.

Given a sample $\boldsymbol{x}$, and a set of $N$ random samples, $X = \{\boldsymbol{x}_1, ..., \boldsymbol{x}_N\}$, which contains one positive sample $\boldsymbol{x}_+$ and $N-1$ negative samples $\boldsymbol{x}_-$, we minimize the following negative log-likelihood:

$$\text{InfoNCE} = -\mathop{\mathbb{E}}_X \left[ \log \frac{e^{\text{sim}(\boldsymbol{x},\boldsymbol{x}_+)/\tau}}{\sum_{\boldsymbol{x}_i \in X} e^{\text{sim}(\boldsymbol{x},\boldsymbol{x}_i)\tau}} \right], \quad (2)$$

where $\tau$ is a temperature hyperparameter. Following SimCSE (Gao et al., 2021), we employ cosine similarity as a measure to compare the representations of positive pairs and negative pairs: $\frac{\boldsymbol{h}_1 \cdot \boldsymbol{h}_2}{\|\boldsymbol{h}_1\|\|\boldsymbol{h}_2\|}$. In this paper, we use pre-trained models such as mBERT (Devlin et al., 2019) and XLMR (Conneau et al., 2019) to encode the input texts and only train all the adapters using the InfoNCE objective.

## 4 The InteMATs Approach

We now introduce InteMATs, which integrates multilingual adapters to enhance the representations of a fixed MLLM. InteMATs involves two stages. In the first stage, we pre-train two multilingual adapters to be specialized in processing texts of different levels of granularity: sentence-level multilingual adapters (MATs-ST) (§4.1) and document-level multilingual adapters (MATs-DT) (§4.2). Inspired by the findings (Park et al., 2023) which

unraveled the complementary properties of CL and masked image modeling objectives, we use CL to train adapters to augment the MLLMs that have been initially trained with the MLM objective. The goal is to enhance the cross-lingual alignment of MLLMs with pluggable adapters while retaining their pre-learned knowledge. In the second stage, we show how to integrate these adapters for cross-lingual transfer tasks. In this paper, we employ AdapterFusion (Pfeiffer et al., 2021) for this purpose (§4.3). InteMATs, however, is not limited to the choice of MLLMs and fusion algorithms. We show the working mechanisms of InteMATs and a concrete example of MATs-ST in Figure 1.

### 4.1 Sentence-level Multilingual Adapters

**Notations.** We designate English as the source language and all the other languages as target languages. To train sentence-level multilingual adapters (MATs-ST), we curate an entity-aligned parallel dataset consisting of $N$ sentences in $J$ languages: $\mathcal{D}_{\text{ST}} = \{D^j\}_{j=1}^J$, where $D^j = \{x_i\}_{i=1}^N$. We employ a pre-trained MLLM as the text encoder and use the hidden states from the penultimate layer of the MLLM as text representations. For example, with mBERT, $x_i$ is encoded into a sequence of $m+1$ token representations: $\boldsymbol{h}_i = <\boldsymbol{e}([\text{CLS}]), \boldsymbol{e}(t_1), ..., \boldsymbol{e}(t_{m-1}), \boldsymbol{e}([\text{SEP}]) >$. Here, [CLS] and [SEP] are classification and separator tokens specially used for learning positional and structural information.

**The pre-training objective.** For every English sample $x^{en}$, we randomly select another English sample $x_+^{en}$ and $N$ non-English samples $x_-^j$ to create contrastive data pairs. We denote the average hidden states of the sequence of tokens as $\boldsymbol{h}_{st}$, and take it as the sentence representation for a given sentence sample. Similarly, we use $\boldsymbol{h}_{et}$ as the representation for the aligned entity.

We train MATs-ST on top of a fixed MLLM by minimizing the following contrastive loss on the sentence-level corpus $\mathcal{D}_{\text{ST}}$:

$$\mathcal{L}_{\text{MATs-ST}}(\boldsymbol{v}_s) = \mathop{\text{InfoNCE}}_{\mathcal{D}_{\text{ST}}}(\boldsymbol{h}_{st}^{en}, \boldsymbol{h}_{st}^j; \boldsymbol{v}_s) \quad (3)$$

$$+ \mathop{\text{InfoNCE}}_{\mathcal{D}_{\text{ST}}}(\boldsymbol{h}_{et}^{en}, \boldsymbol{h}_{et}^j; \boldsymbol{v}_s), \quad (4)$$

where the superscript $en$ represents English and $j \in [1, J]$ represents one of the $J$ languages. $\boldsymbol{v}_s$ denotes the parameters of MATs-ST. We encourage cross-lingual alignment at both sentence-level and entity-level representations.

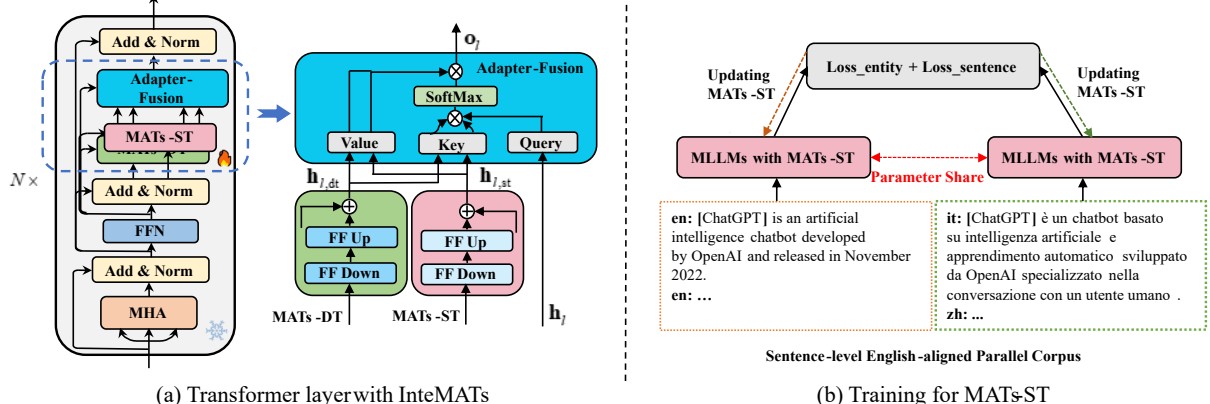

(a) Transformer layer with InteMATs

(b) Training for MATs-ST

**Figure 1: (a):** The structure of InteMATs in each transformer layer. It includes two multilingual adapters (i.e., MATs-ST and MATs-DT) and an adapter-fusion module. The fusion module learns the *Key*, *Value*, *Query* matrices to fuse two pre-trained adapters. MATs-ST and MATs-DT are fixed when updating the fusion module. **MHA** means multi-heads attention mechanism. **Add&Norm** means addition and layer normalization operation. **FFN** means the feed-forward neural network. **(b):** An example of training MATs-ST on the sentence-level parallel corpus. The MLLM backbone is fixed during training.

## 4.2 Document-level Multilingual Adapter

As Hu et al. (2020) demonstrates, the cross-lingual transfer performance of MLLMs tends to degenerate on longer texts, potentially due to their limited capability of understanding longer contexts. Motivated by this observation, we specially train document-level multilingual adapters (MATs-DT) for MLLMs. Similar to the pre-training of MATs-ST, we curate a document-level parallel dataset comprising $J$ languages: $\mathcal{D}_{\mathrm{DT}} = \{D^j\}_{j=1}^J$. For each document $d$ in $D^j$, we also average the hidden states from the penultimate layer of an MLLM as the document-level representation: $\boldsymbol{h}_{dt}$.

**The pre-training objective.** The pre-training setup of MATs-DT is similar to that of MATs-ST except using longer input texts. We encourage adapters to capture cross-lingual alignment on the document-level corpus, $\mathcal{D}_{\mathrm{DT}}$, by minimizing the following contrastive loss:

$$\mathcal{L}_{\mathrm{MATs-DT}}(\boldsymbol{v}_d) = \underset{\mathcal{D}_{\mathrm{DT}}}{\mathrm{InfoNCE}}(\boldsymbol{h}_{dt}^{en}, \boldsymbol{h}_{dt}^{j}; \boldsymbol{v}_d), \quad (5)$$

where $\boldsymbol{v}_d$ denotes the parameters of MATs-DT. Note that we do not explicitly include the loss for entity-level alignment as Eq.4. Our experiments show that incorporating such constraint does not improve cross-lingual transfer performance and, in fact, leads to a decrease in performance. This suggests a conflict between entity-level alignment, which focuses on capturing local information, and document-level alignment, which emphasizes capturing global information.

## 4.3 Integrating Multilingual Adapters

We now introduce the second stage: knowledge composition. Inspired by AdapterFusion (Pfeiffer et al., 2021), which trains multiple task-specific adapters on top of a shared pre-trained model. We propose to train InteMATs in a similar way to fuse the multilingual adapters on cross-lingual tasks. By incorporating both MATs-ST and MATs-DT at each transformer layer, we seek a more comprehensive view for capturing transferrable features from varying lengths of context.

As illustrated in Figure 1, we append the AdapterFusion module right after the MATs-ST and MATs-DT modules, followed by a residual connection to the original transformer output, $\boldsymbol{h}_l$, at layer $l$. The outputs of MATs-ST ($\boldsymbol{h}_{st,l}$) and MATs-DT ($\boldsymbol{h}_{dt,l}$) are used as inputs for both the *Value* and *Key* transformations, and the new hidden states after fusion are as follows:

$$Q_l = \boldsymbol{h}_l W_q, Q_l \in \mathbb{R}^{1 \times d} \quad (6)$$

$$K_l = [\boldsymbol{h}_{st,l} W_k, \boldsymbol{h}_{dt,l} W_k], K_l \in \mathbb{R}^{2 \times d} \quad (7)$$

$$V_l = [\boldsymbol{h}_{st,l} W_v, \boldsymbol{h}_{dt,l} W_v], V_l \in \mathbb{R}^{2 \times d} \quad (8)$$

$$O_l = \mathrm{softmax}(Q_l K_l^{\top} / \sqrt{d}) V_l, O_l \in \mathbb{R}^{1 \times d} \quad (9)$$

where $d$ represents the size of hidden states, $[\cdot, \cdot]$ indicates the concatenation of vectors, and $W_q, W_k, W_v$ are $d \times d$ matrices for computing cross attentions (Vaswani et al., 2017).

We train InteMATs on each cross-lingual task to determine the task-specific importance weights for MATs-ST and MATs-DT. In this process, the

parameters of MATs-ST and MATs-DT are fixed while the parameters of the fusion module, $\boldsymbol{v}_f$, are optimized by minimizing, e.g., cross-entropy loss, on each task-specific dataset $\mathcal{D}_T$:

$$\mathcal{L}_{\text{InteMATs}}(\boldsymbol{v}_f) = \mathbb{E}_{(\boldsymbol{x},y)\in\mathcal{D}_T} -\log P(\psi_{\boldsymbol{v}_f}(\boldsymbol{x}) = y) \quad (10)$$

# 5 Experiments

In this section, we present the evaluation details and results of InteMATs across sentence-level and document-level cross-lingual tasks.

## 5.1 Experimental Setup

**Pre-training Corpus.** We collect a large, entity-aligned, multilingual dataset from Wikipedia[2]. Each data sample is a summary text with a maximum length of 384 to describe an entity, which spans multiple languages. The dataset covers 42 languages in total, including those extensively studied in the popular XTREME benchmark (Hu et al., 2020). We treat English as the source language and ensure each English sample has at least four parallel samples from other languages. We utilize the first sentence of each summary text and curate the sentence-level parallel dataset $\mathcal{D}_{\text{ST}}$ for training MATs-ST and use the entire raw text $\mathcal{D}_{\text{DT}}$ for training MATs-DT. More details about the dataset can be found in Appendix A.1.

**MLLM Backbones.** We experiment on three representative MLLMs, the base version of mBERT (Devlin et al., 2019), both the base and large versions of XLMR (Conneau et al., 2019), to show the effects on different model types and model scales. Hyperparameters setup under each MLLM can be found in Appendix A.1.

## 5.2 Baselines

We compare InteMATs against the following 12 baselines. 1) MLLMs without adapters: we directly fine-tune the backbone MLLMs, mBERT, and XLMR (base and large versions), on downstream cross-lingual tasks. We also compare with another four SOTA MLLMs including VECO (Luo et al., 2021), XLM-ALIGN (Chi et al., 2021b), KMLM (Liu et al., 2022), and XLM-E (Chi et al., 2022). 2) On sentence retrieval tasks, we compare with two SOTA methods: mSimCSE (Wang et al., 2022) and InfoXLM (Chi et al., 2021a), which fine-tune the entire MLLM on their pre-training corpus before applying to downstream tasks. 3) On RELX

---
[2]https://github.com/martin-majlis/Wikipedia-API

dataset (Köksal and Özgür, 2020), we compare with MTMB (Köksal and Özgür, 2020) which introduces distant supervision to enhance MLLMs, and MLKG (Hou et al., 2022) which incorporates multilingual knowledge graphs into MLLMs. 4) On XQuAD dataset (Artetxe et al., 2020a), we compare with two adapter-based methods: MLKG (Hou et al., 2022) and MAD-X (Pfeiffer et al., 2020) which uses one task-specific adapter. Note that we only report the results based on large version of XLMR for MLKG and KMLM.

**Evaluation Setup.** We divide the languages in each downstream task into two types: **Sup.**: stands for *supervised language*, which is used for fine-tuning, and **ZS**: stands for *zero-shot language*, which is only used for testing.

## 5.3 Cross-lingual Semantic Textual Similarity

We first evaluate the models on the Cross-lingual Semantic Textual Similarity (STS) task (Cer et al., 2017) to assess the quality of their representations in capturing universal semantics. Following mSim-CSE (Wang et al., 2022), we average the embeddings from the first and last layers.

Table 1 presents the semantic textual similarity among the Arabic (ar), Spanish (es), English (en), and Turkish (tr) languages. Comparing InteMATs (XLMR) with the SOTA model, mSimCSE, we observe that on *monolingual* groups (ar→ar, es→es), mSimCSE demonstrates the highest textual similarity, while on *cross-lingual* groups (ar→en, es→en, tr→en), InteMATs produces the highest textual similarity. Note that mSimCSE employs the large version of XLMR as the backbone during pre-training. This implies that InteMATs can enhance the representations of XLMR in capturing *cross-lingual semantics*. Moreover, mSimCSE requires fine-tuning the entire MLLM in the pre-training stage while InteMATs only trains a set of adapters (See Appendix 5.11.). However, when conditioned on mBERT, it does not bring improvement, showing the limit on the choice of MLLMs.

**Table 1:** Spearman rank correlation ($\rho$) on the cross-lingual semantic textual similarity task (higher, better).

| Models | ar->ar | ar->en | es->es | es->en | tr->en | Avg. |
|---|---|---|---|---|---|---|
| mSimCSE | **72.3** | 48.4 | **83.7** | 57.6 | 53.4 | 63.1 |
| mBERT | 55.2 | 28.3 | 68.0 | 23.6 | 17.3 | 38.5 |
| InteMATs (mBERT) | 53.4 | 26.0 | 66.0 | 21.4 | 15.9 | 36.5 |
| XLMR$_{base}$ | 44.0 | 18.8 | 65.1 | 6.8 | 6.5 | 28.2 |
| InteMATs (XLMR$_{base}$) | 47.8 | 33.8 | 74.6 | 42.2 | 43.3 | 48.3 |
| XLMR$_{large}$ | 53.6 | 26.2 | 68.1 | 10.7 | 10.5 | 33.8 |
| InteMATs (XLMR$_{large}$) | 65.7 | **58.2** | 73.2 | **60.9** | **59.2** | **63.4** |

## 5.4 Cross-lingual Sentence Retrieval

We use BUCC (Zweigenbaum et al., 2017) and Tatoeba (Artetxe and Schwenk, 2019a) datasets to evaluate the cross-lingual sentence retrieval performance of InteMATs. Specifically, given a sample from the source language, e.g., English, the model should correctly retrieve all the similar samples from other xx languages, and vice versa. On Tatoeba, we follow XLM-E (Chi et al., 2022) and mSimCSE (Wang et al., 2022) and report the results on both the 14 common languages (Tatoeba-14) and all the 36 languages (Tatoeba-36).

Table 2 presents the overall performance comparison. InteMATs, when conditioned on XLMR$_{large}$, outperforms all the baselines by a large margin, establishing a new state-of-the-art on the unsupervised settings of BUCC and Tatoeba. On all selected MLLMs, InteMATs outperforms its counterparts regardless of the model scale, indicating that there is significant potential for enhancing cross-lingual alignment in the representations provided by pre-trained MLLMs. InteMATs outperforms the second-best state-of-the-art model, mSimCSE, implying that pre-training adapters through CL is better than the traditional full-model fine-tuning approach in aligning cross-lingual representations. Moreover, InteMATs outperforms other competitive MLLMs, namely XLM-E and InfoXLM, which employ the same backbone but with distinct pre-training tasks.

**Table 2:** The cross-lingual sentence retrieval performance. We report the average F1 score of four languages for BUCC and the average accuracy@1 score for Tatoeba.

| Models | BUCC-4 | Tatoeba-14 | | Tatoeba-36 | |
|---|---|---|---|---|---|
| | xx->en | en->xx | xx->en | en->xx | xx->en |
| mSimCSE | 87.5 | - | 82.0 | - | 78.0 |
| INFOXLM | - | 80.6 | 77.8 | 68.6 | 67.3 |
| XLM-E | - | 74.4 | 72.3 | 65.0 | 62.3 |
| mBERT | 56.7 | 48.8 | 57.2 | 34.9 | 34.3 |
| InteMATs (mBERT) | 68.5 | 53.7 | 59.3 | 37.0 | 34.9 |
| XLMR$_{base}$ | 76.2 | 67.7 | 71.6 | 29.7 | 67.7 |
| InteMATs (XLMR$_{base}$) | 80.8 | 82.5 | 81.1 | 73.8 | 71.8 |
| XLMR$_{large}$ | 76.7 | 65.8 | 42.5 | 53.5 | 39.7 |
| InteMATs (XLMR$_{large}$) | **91.5** | **92.3** | **90.9** | **88.8** | **82.0** |

## 5.5 Cross-lingual Sentence Classification

We conduct performance evaluation on RELX (Köksal and Özgür, 2020), the cross-lingual relation extraction dataset, and PAWS-X (Zhang et al., 2019), the paraphrase detection dataset. RELX contains five languages and PAWS-X contains seven languages. We follow the XTREM benchmark (Hu et al., 2020) and train adapters only on English data and test on non-English data. Following MLKG (Hou et al., 2022), we report the performance on the source language, Sup(en), the average performance on all the other $n$ zero-shot languages, ZS($n$), and the average performance on all languages in the dataset.

Table 3 presents the performance comparison results. Generally, InteMATs still takes the highest spot on both datasets, especially on zero-shot languages (ZS). Interestingly, the average performance of InteMATs matches that of VECO on PAWS-X. However, VECO requires fine-tuning the entire XLMR model and is supported with a huge pre-training corpus covering 50 languages (Luo et al., 2021) while InteMATs only trains a few pluggable adapters. Compared with its backbone models, mBERT and XLMR, InteMATs consistently enhances their cross-lingual zero-shot performance by $2\% \sim 3.2\%$. However, the performance gains on the source language (en) are not consistent, indicating that InteMATs may focus on enhancing the cross-lingual alignment on target languages.

**Table 3:** The cross-lingual sentence classification performance. We report F1 score for RELX and Accuracy (Acc.) for PAWS-X. Results of VECO, XLM-E, XLM-ALIGN, KMLM, MTMB, and MLKG are taken from their original papers.

| Models | RELX | | | PAWS-X | | |
|---|---|---|---|---|---|---|
| | Sup.(en) | ZS (4) | Avg. | Sup.(en) | ZS (6) | Avg. |
| VECO | - | - | - | 96.2 | 87.4 | 88.7 |
| XLM-ALIGN | - | - | - | 95.1 | 85.4 | 86.8 |
| XLM-E | - | - | - | - | - | 87.1 |
| KMLM | - | - | - | - | - | 88.0 |
| MTMB | 63.6 | 59.6 | 60.4 | - | - | - |
| MLKG | **64.1** | 60.4 | 61.1 | - | - | - |
| mBERT | 61.8 | 57.4 | 58.3 | 94.0 | 80.0 | 81.9 |
| InteMATs (mBERT) | 63.7 | 59.4 | 60.2 | 94.9 | 83.2 | 84.9 |
| XLMR$_{large}$ | 63.1 | 59.1 | 59.9 | 94.7 | 85.0 | 86.4 |
| InteMATs (XLMR$_{large}$) | 61.1 | **61.7** | **61.7** | 95.7 | **87.5** | **88.7** |

## 5.6 Cross-lingual Syntactic Analysis

We conduct evaluations on a cross-lingual Part-of-Speech (POS) dataset (Nivre et al., 2020) from the XTREME benchmark (Hu et al., 2020) to assess a model's capability of capturing syntactic structures and grammatical properties across 33 languages. All models are trained on English data.

As shown in Table 4, InteMATs achieves comparable performance to XLM-ALIGN and outperforms VECO and XLM-E on average. This suggests that instead of fine-tuning the entire MLLMs, transferring the syntactic knowledge acquired from English data to other languages can be easily achieved by only tuning a few adapters. It holds true regardless of the choice of MLLMs. Moreover, InteMATs achieves more performance gains for zeros-hot languages, showing the advantage of adapter tuning for MLLMs.

**Table 4:** The cross-lingual structure prediction performance. Accuracy (Acc.) is used for POS evaluation.

| Models | POS | | |
|---|---|---|---|
| | Sup.(en) | ZS (32) | Avg. |
| VECO | 95.9 | 74.5 | 75.1 |
| XLM-ALIGN | 95.9 | 75.4 | **76.0** |
| XLM-E | - | - | 74.2 |
| KMLM | - | - | 72.8 |
| mBERT | 95.5 | 70.8 | 71.5 |
| InteMATs (mBERT) | 95.3 | 71.8 | 72.5 |
| XLMR$_{base}$ | 95.9 | 72.2 | 72.9 |
| InteMATs (XLMR$_{base}$) | 96.0 | 72.4 | 73.1 |
| XLMR$_{large}$ | 96.1 | 73.1 | 73.8 |
| InteMATs (XLMR$_{large}$) | **96.1** | **74.9** | 75.6 |

## 5.7 Cross-lingual Question-Answering

Question-answering (QA) requires a model to understand a given long context so as to correctly answer the questions by extracting the text span for the true answer from the context. We conduct evaluations on two popular multilingual QA benchmarks: XQuAD (Artetxe et al., 2020a) and TydiQA (Clark et al., 2020).

Table 5 presents the performance comparison between InteMATs and SOTA models. In general, InteMATs excels all the full-model fine-tuning baselines (VECO, XLM-E, XLM-ALIGN, and KMLM), and adapter-based baselines (MLKG and MAD-X). Conditioned on the same large version of XLMR, InteMATs surpasses the second-best baseline, KMLM, by $0.3\%$ on XQuAD and $3.2\%$ on TydiQA, at a low cost of training. On both supervised and zero-shot benchmarks, InteMATs consistently outperforms fine-tuning the backbone models, mBERT and XLMR, indicating the advantage of incorporating multilingual adapters into pre-trained MLLMs.

**Table 5:** The cross-lingual question-answering performance. We report their F1 score and Exact Matching (F1/EM) scores.

| Models | XQuAD | | | TydiQA | | |
|---|---|---|---|---|---|---|
| | Sup.(en) | ZS (10) | Avg. | Sup.(en) | ZS (8) | Avg. |
| VECO | 87.6/76.5 | 76.3/60.3 | 77.3/61.8 | 71.3/58.2 | 67.1/48.0 | 67.6/49.1 |
| XLM-ALIGN | 85.7/74.6 | 73.6/57.4 | 74.7/59.0 | 69.4/56.2 | 61.2/43.4 | 62.1/44.8 |
| XLM-E | -/- | -/- | 74.3/58.2 | -/- | -/- | 57.8/40.6 |
| KMLM | -/- | -/- | 77.3/61.7 | -/- | -/- | 67.9/50.4 |
| MLKG | **88.0/77.6** | 75.7/59.7 | 76.8/61.3 | -/- | -/- | -/- |
| MAD-X | 84.7/72.6 | 69.7/54.0 | 71.1/55.7 | -/- | -/- | -/- |
| mBERT | 83.5/72.2 | 62.6/47.2 | 64.5/49.4 | 75.3/63.6 | 57.7/41.5 | 59.7/43.9 |
| InteMATs (mBERT) | 85.9/73.9 | 63.5/48.0 | 65.5/50.4 | **75.6/63.1** | 58.2/42.3 | 60.2/44.6 |
| XLMR$_{base}$ | 83.4/72.7 | 70.1/54.6 | 68.8/52.8 | 56.8/43.6 | 42.1/24.5 | 43.8/26.6 |
| InteMATs (XLMR$_{base}$) | 83.3/71.9 | 70.7/55.5 | 69.5/53.9 | 60.5/47.3 | 46.8/29.7 | 48.3/31.7 |
| XLMR$_{large}$ | 86.5/75.7 | 75.6/59.3 | 76.6/60.8 | 71.5/56.8 | 64.3/43.5 | 65.1/45.0 |
| InteMATs (XLMR$_{large}$) | 87.4/76.4 | **76.6/60.6** | **77.6/62.0** | 73.8/61.3 | **70.8/52.7** | **71.1/53.7** |

## 5.8 Closing the Cross-Lingual Transfer Gap

We summarize the above cross-lingual performance results and conclude that InteMATs can effectively mitigate the issue of cross-lingual performance degeneration in pre-trained MLLMs, as shown in Table 6. All models are trained using the source English language, achieving a perfor-

mance of $S_{en}$. They are subsequently tested on other languages, yielding an average performance of $S_{tgt}$. The performance gap, $\Delta = S_{en} - S_{tgt}$, serves as an evaluation metric to assess the degree of cross-lingual transfer degradation.

On all but the POS task, InteMATs based on XLMR$_{large}$, demonstrates the lowest cross-lingual performance degeneration, highlighting its advantage in enhancing cross-lingual knowledge transfer.

**Table 6:** The performance gap ($\Delta$) between the source language and target languages on cross-lingual transfer tasks. A lower score indicates better cross-lingual transferability.

| Model | PAWS-X (Acc.) | RELX (F1) | POS (Acc.) | XQuAD (F1\EM) | TydiQA (F1\EM) |
|---|---|---|---|---|---|
| mBERT | 14.0 | 4.4 | 24.7 | 20.9/25.0 | 17.6/22.1 |
| InteMATs (mBERT) | 11.7 | 4.3 | 23.5 | 22.4/25.9 | 17.4/20.8 |
| XLMR | 9.7 | 4.0 | 23 | 10.9/16.4 | 7.2/16.4 |
| VECO | 8.8 | - | 21.4 | 11.3/16.4 | 4.2/10.2 |
| XLM-ALIGN | 9.7 | - | **20.5** | 12.1/17.2 | 8.2/12.8 |
| InteMATs (XLMR$_{large}$) | **8.2** | **-0.6** | 21.2 | **10.8/15.8** | **3.0/8.6** |

## 5.9 Scaling to Low-resource Languages

We further study how InteMATs performs on languages out of our pre-training corpus. We use eight low-resource languages from the Tatoeba dataset: Cantonese (yue), Vietnamese (vie), Tagalog (tgl), Irish (gle), Georgian (kat), Khmer (khm), Telugu (tel), Serbian (srp).

Table 7 presents the results. We find that directly applying MLLMs on low-resource languages yields poor performance, with an average accuracy of approximately $25\% \sim 35\%$. In contrast, when conditioned on XLMR$_{large}$, InteMATs significantly improves the performance to $62\%$. These findings confirm our hypothesis that pre-trained MLLMs exhibit poor cross-lingual alignment on low-resource languages due to their scarcity of training data during pre-training. InteMATs effectively enhances MLLMs by capturing better cross-lingual alignment information, enabling generalization to unseen low-resource languages. Appendix A.2 provides more details.

**Table 7:** The performance on the low-resource languages from Tatoeba dataset for cross-lingual retrieval accuracy (Acc.).

| Models | yue | vi | tl | ga | ka | km | te | sr | Avg. |
|---|---|---|---|---|---|---|---|---|---|
| mBERT | 26.1 | 59.4 | 15.8 | 10.5 | 20.2 | 0.3 | 17.1 | 43.7 | 24.1 |
| InteMATs (mBERT) | 26.8 | 61.0 | 16.0 | 11.8 | 19.6 | 0.4 | 14.1 | 44.6 | 24.3 |
| XLMR$_{base}$ | 24.7 | 67.7 | 29.7 | 16.7 | 37.2 | 19.8 | 30.8 | 58.6 | 35.7 |
| InteMATs (XLMR$_{base}$) | 38.5 | 81.7 | 42.9 | 25.5 | 53.4 | 38.5 | 51.7 | 73.0 | 50.7 |
| XLMR$_{large}$ | 21.3 | 41.8 | 18.9 | 9.2 | 15.9 | 15.7 | 26.0 | 47.4 | 24.5 |
| InteMATs (XLMR$_{large}$) | 52.5 | 91.6 | 51.1 | 31.2 | 68.8 | 48.9 | 70.9 | 83.6 | 62.3 |

## 5.10 Ablation Study and Analysis

We conduct ablation studies on six tasks to unravel the individual impact of each component in

InteMATs. Specifically, we compare the backbone MLLMs, MATs-ST, MATs-DT, and InteMATs. The ablation results in Table 8 show that MATs-ST generally outperforms MATs-DT on sentence-level tasks, while MATs-DT performs better than MATs-ST on document-level tasks. Both MATs-ST and MATs-DT outperform the backbone MLLMs, particularly with substantial gains on BUCC, Tatoeba, and TydiQA. By effectively incorporating these two modules, InteMATs achieves the best performance across the benchmarks.

**Table 8:** The ablation results for both MLLMs on various cross-lingual language tasks.

| Models | BUCC (F1) | Tatoeba (Acc.) | RELX (F1) | PAWS-X (Acc.) | XQuAD (F1\EM) | TydiQA (F1\EM) |
|---|---|---|---|---|---|---|
| | | | mBERT backbone | | | |
| mBERT | 56.7 | 34.6 | 58.3 | 81.9 | 64.5/49.4 | 59.7/43.9 |
| MATs-ST | 68.0 | 34.3 | 59.9 | 83.4 | 65.2/49.8 | 58.1/42.0 |
| MATs-DT | 67.5 | 35.4 | 58.7 | 84.5 | 65.4/50.7 | 59.7/43.6 |
| InteMATs | **68.5** | **36.0** | 60.2 | 84.9 | 65.5/50.4 | 60.2/44.6 |
| | | | XLMR$_{large}$ backbone | | | |
| XLMR$_{large}$ | 76.7 | 46.6 | 59.9 | 86.4 | 76.6/60.8 | 65.1/45.0 |
| MATs-ST | 91.3 | 84.3 | 60.9 | 88.4 | 77.3/61.4 | 70.1/51.8 |
| MATs-DT | 87.6 | 74.1 | 59.9 | 88.0 | 77.5/62.1 | 70.9/53.3 |
| InteMATs | **91.5** | **85.4** | **61.7** | **88.7** | **77.6/62.0** | **71.1/53.7** |

### 5.10.1 Analysis of Pre-training Corpus

We further study whether our proposed different granularities of pre-training corpus boosts MLLMs' performance on cross-lingual understanding tasks. We compare with Initialized (adding a randomly initialized adapter without pre-training) and Mixed (adding an adapter trained on a non-distinguishing granularity of the pre-training corpus), as shown in Table 9.

**Table 9:** The performance on various cross-lingual tasks under the MLLMs pre-trained on different pre-training corpus.

| Model | PAWS-X (Acc.) | Tatoeba (Acc.) | RELX (F1) | XQuAD (F1\EM) | TydiQA (F1\EM) |
|---|---|---|---|---|---|
| | | | mBERT backbone | | |
| Initialized | 83.6 | 34.1 | 57.9 | **65.4/50.4** | 55.8/43.7 |
| Mixed | 84.2 | **37.6** | 56.4 | 64.4/49.3 | 53.5/38.6 |
| InteMATs | **84.9** | 36.0 | **60.2** | 65.5/50.0 | **60.2/44.6** |
| | | | XLMR$_{large}$ backbone | | |
| Initialized | - | 65.8 | 60.7 | 77.4/62.0 | 69.8/53.1 |
| Mixed | 86.0 | 73.9 | 57.4 | 77.2/61.8 | 67.8/49.6 |
| InteMATs | **88.7** | **85.4** | **61.7** | **77.6/62.0** | **71.1/53.7** |

We find that InteMATs consistently outperform Initialized across all tasks, with an average improvement of 2% on mBERT and 4.2% on XLMR backbones. Meanwhile, InteMATs surpass Mixed on all tasks except Tatoeba, achieving average gains of 2.1% on mBERT and 4.4% on XLMR backbones. These findings emphasize that InteMATs by pre-training individually on more diverse range of text granularities can precisely capture and integrate different text granularities of cross-lingual

alignments, transferring this alignment knowledge to understanding tasks and yielding performance improvements.

### 5.10.2 Analysis of Pre-training Tasks

We evaluate the pre-training tasks of CL's advantage in capturing global cross-lingual alignment information. We compare with mBERT-FT, which is fully fine-tuned on the entire raw text $\mathcal{D}_{DT}$, and InteMATs-MLM, which uses the pre-training task of MLM instead of CL to train each adapters.

Table 10 shows the results. We can observe that MLLMs with adapters can achieve performance enhancements on all tasks but Tatoeba. This suggests employing external adapters for training is more effective for incremental cross-lingual knowledge's learning, than full model fine-tuning. Meanwhile, InteMATs excel over other MLLMs on four tasks, showing average gains of 1.6% over InteMATs-MLM, 2% over Initialized, and 3% over mBERT-FT. These results underscore that applying the pre-training task of CL to train adapter can enhance global cross-lingual alignment information, enabling knowledge transferring to understanding tasks and boosting performance.

**Table 10:** The performance on various cross-lingual tasks under different pre-training tasks.

| Model | POS (Acc.) | Tatoeba (Acc.) | BUCC (F1) | TydiQA (F1\EM) |
|---|---|---|---|---|
| mBERT-FT | 70.3 | **37.2** | 64.1 | 54.8/40.5 |
| Initialized | 71.3 | 34.1 | 67.1 | 57.9/40.6 |
| InteMATs-MLM | 71.4 | 35.7 | **68.5** | 56.0/40.6 |
| InteMATs | **72.5** | 36.0 | **68.5** | **60.2/44.6** |

### 5.10.3 Analysis of Fusion Activation

InteMATs learns to fuse knowledge from MATs-ST and MATs-DT for different tasks. We retrieve the activation values of its fusion module and visualize the weight distributions for four tasks in Figure 2 (POS, RELX, XQuAD, and TydiQA), with increasing input text lengths.

We observe a consistent pattern in InteMATs, where it favors MATs-ST for sentence-level tasks and MATs-DT for document-level tasks. Specifically, on POS and RELX tasks, it relies more on the representations from MATs-ST, while on XQuAD and TydiQA, it relies more on MATs-DT. Moreover, as the network goes deeper, the degree of reliance on MATs-DT increases. This finding confirms with our intuition that granularity-specific adapters are specialized in handling texts of varying lengths. As a result, InteMATs can effectively

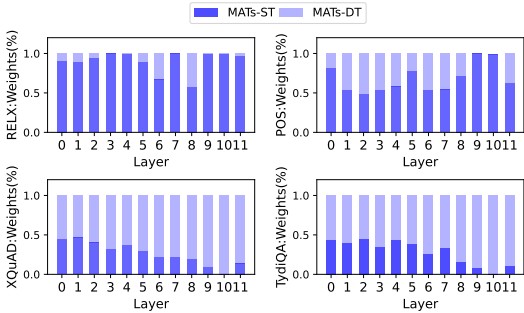

**Figure 2:** InteMATs activation at different layers. We follow the setting from Pfeiffer et al. (2021) to calculate the SoftMax activation for the two adapters, MATs-ST and MATs-DT.

leverage these adapters to enhance cross-lingual alignment regardless of the specific task at hand.

### 5.10.4 Layer-wise Representation Analysis

We examine InteMATs layer by layer to unravel on which layers it enhances cross-lingual transfer performance. Figure 3 compares InteMATs and XLMR in both base and large versions. We report the sentence retrieval accuracy on the Tatoeba dataset (Artetxe and Schwenk, 2019a) using representations from each transformer layer.

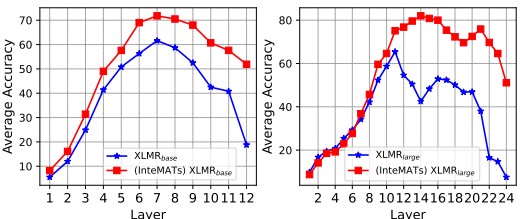

**Figure 3:** Average accuracy on 36 language pairs from Tatoeba dataset in the xx->en directions.

We find that InteMATs achieves similar performance to XLMR in the early layers. However, in later layers (layer 2 onwards for the base version and layer 9 onwards for the large version), InteMATs significantly outperforms XLMR. This comparison reveals that the cross-lingual transfer capability of InteMATs is gradually developed in the later layers, which provides improved cross-lingual alignment. This finding complies with previous research that later layers of Transformers tend to extract high-level features (Clark et al., 2019).

### 5.11 Model Configuration

We compare the pre-training budget for enhancing the cross-lingual alignments against existing MLLMs in Table 11. The results reveal that InteMATs offers better parameter efficiency during pre-training and performance improvements on

cross-lingual understanding tasks across various text granularities, requiring fewer computational parameters and smaller training corpus.

**Table 11:** The implementation details of the existing MLLMs and InteMATs.

| Model | #Pre-trained Parameters | #Languages | Data Size |
|---|---|---|---|
| MMTE | 375M | 103 | - |
| XLM-E | 840M | 100 | 2,500GB |
| VECO | 662M | 50 | 1,360GB |
| XLM-ALING | 279M | 94 | 2,187GB |
| MLKG (XLMR$_{large}$) | 20M | 84 | 36GB |
| InteMATs (mBERT) | 3M | 42 | 2.1GB |
| InteMATs (XLMR$_{large}$) | 13M | 42 | 2.1GB |

## 6 Conclusion

In this paper, we introduce InteMATs, a novel approach that integrates granularity-specific multilingual adapters, including sentence-level multilingual adapters (MATs-ST) and document-level multilingual adapters (MATs-DT), to enhance MLLMs in cross-lingual transfer tasks. On top of a fixed MLLM (e.g., mBERT and XLMR), we train these adapters on our curated parallel corpus using the contrastive learning objective. Our experiments demonstrate that InteMATs significantly enhanced the cross-lingual transfer performance of MLLMs across sentence-level and document-level tasks. Our comprehensive analyses show that InteMATs can automatically leverage corresponding adapters when dealing with different kinds of tasks.

## Limitations

We identify a few limitations of our current work.

- First, InteMATs demonstrates limited improvements on structure prediction tasks, i.e., POS dataset. This is not surprising as syntactic structures are not universal across different languages. However, it is possible to share knowledge between languages from the same family, e,g., Romance languages (es, pt, it, fr, ro). We encourage future researchers to pay more attention to the syntactic cross-lingual alignment for MLLMs.

- Second, the publicly available benchmarks for cross-lingual transfer evaluation are dominated by sentence-level tasks. As a result, performance comparisons on existing benchmarks could be inadequate to demonstrate a model's capability of handling longer contexts and transfer that ability to different languages.

A more comprehensive cross-lingual transfer benchmark is needed.

## Reproducibility Statement

We elaborate the experiment settings and hyperparameters in the Appendix A.1. We will publish our collected parallel datasets soon, as well as our code.

## Ethics Statement

All procedures performed in this work were in accordance with ethical standards. We don't have any ethical concerns in this work.

## Acknowledgements

We extend our gratitude to the anonymous reviewers for their valuable feedback and insights. This work was completed at NTU Singapore and was supported by the China Scholarship Council. Funding was also provided by the National Key R and D Program of China (Grant No.2017YFB1302400) and Shandong Province Agricultural Major Application Technology Innovation Project (Grant No.SD2019NJ014).

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

## A  Appendix

### A.1  Training Details

**Training Data Statistics.**  We present the details of the collected parallel training corpus in Table 12, which covers 42 diverse of languages. Especially, for $\mathcal{D}_{\mathrm{DT}}$, we filter out summaries whose tokenized input sequence lengths are greater than 384. For both $\mathcal{D}_{\mathrm{ST}}$ and $\mathcal{D}_{\mathrm{DT}}$, we split the corpus into training and validation data in a $9:1$ ratio.

**Training Settings.**  Experiments are conducted on 4 Tesla $V100$ GPUs. We set the maximum input sequence length to 384, with a batch size of 8 and an accumulation step of 2. The learning rate is set to $5e-5$, and we use the Adam optimizer with a warm-up step of $1e4$ during training and the random seed is set as 123. Since we adopt the random sampling method and contrastive learning, i.e., we randomly sample aligned text in different languages as positive samples, we only take several 12 hours for training MATs-ST, and 2.5 days at more for training MATs-DT. Both training epochs for MATs-ST and MATs-DT are set as 5 epochs.

### A.2  Scaling to Unseen Languages

Since we adopt contrastive learning loss for multilingual adapters training, aiming to enhance crosslingual alignments by learning the universal features across languages. To assess the generalization to *unseen languages* of InteMATs, we collected two groups of parallel sample sets: sentence-level corpus and document-level corpus. Both different level of corpus involve two language set: *In-Domain set*, including languages in pre-training corpus, and *Un-Domain set*, including languages out-of pre-training corpus [3].

We employ the cosine similarity to assess the cross-lingual alignments of individual adapters (MATs-ST and MATs-DT) and InteMATs on both sentence-level and document-level corpus, and mBERT base version is selected as backbone models. The results are shown in Figure 4. Compared to the vanilla mBERT, the mBERT with any multilingual adapters all consistently and significantly enhance the similarities of aligned embeddings across different text granularities in both the *In-Domain set* and *Un-Domain set*, while InteMATs demonstrate the highest performance. This indicates that

our adapters effectively facilitate mBERT in capturing global representation patterns across languages, thereby enabling substantial cross-lingual enhancements.

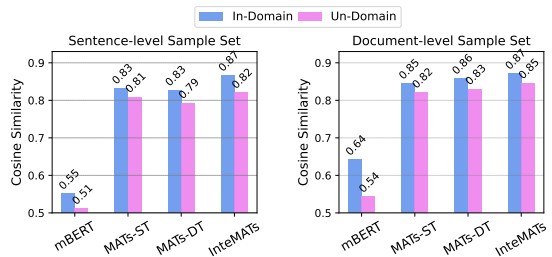

**Figure 4:** The cosine similarities of aligned embeddings of two parallel sample sets.

### A.3  Detailed Experimental Results

We provide detailed results for each language on the cross-lingual language tasks. Specifically, we present the results for cross-lingual sentence-level retrieval benchmarks in Tables 13 (based on BUCC dataset), while the results for the cross-lingual relation extraction and classification benchmark are displayed in Tables 14 (on RELX dataset) and 15 (on PAWS-X dataset). The results for cross-lingual structure prediction are shown in Table 16. Meanwhile, we present the results for cross-lingual question answering tasks in Table 17 (on XQuAD dataset) and 18 (on TydiQA dataset).

---

[3]The In-Domain set primarily utilizes the validation set of the collected corpus A.1. We collect additional parallel data to form an Un-Domain set, which includes languages such as dv, gl, ha, ie, ka, ps, ro, sd, sr, tg, tt, wuu, yue.

**Table 12:** Statistics about the parallel training corpus in different languages. Meanwhile, IE means Indo-European, which has the most members. The language selection aims to cover a wide range of language tasks.

| Language | ISO Code | Language Family | # of Instances |
|---|---|---|---|
| English | en | IE: Germanic | 216920 |
| Spanish | es | IE: Romance | 98813 |
| Arabic | ar | Afro-Asiatic | 20366 |
| Bengali | bn | IE: Indo-Aryan | 4258 |
| Greek | el | IE: Hellenic | 4541 |
| Danish | da | IE: Germanic | 36151 |
| German | de | IE: Germanic | 100601 |
| French | fr | IE: Romance | 108312 |
| Hebrew | he | Afro-Asiatic | 7453 |
| Hungarian | hu | Uralic | 29473 |
| Indonesian | id | Austronesian | 15677 |
| Italian | it | IE: Romance | 79465 |
| Japanese | ja | Japonic | 3789 |
| Korean | ko | Koreanic | 4784 |
| Dutch | nl | IE: Germanic | 89033 |
| Malay | ms | Austronesian | 9480 |
| Portuguese | pt | IE: Romance | 50019 |
| Polish | pl | IE: Slavic | 46420 |
| Russian | ru | IE: Slavic | 39592 |
| Swedish | sv | IE: Germanic | 42197 |
| Telugu | te | Dravidian | 1065 |
| Tamil | ta | Dravidian | 992 |
| Thai | th | Kra-Dai | 1494 |
| Turkish | tr | Turkic | 12005 |
| Ukrainian | uk | IE: Slavic | 19650 |
| Vietnamese | vi | Austroasiatic | 15880 |
| Mandarin | zh | Sino-Tibetan | 1855 |
| Basque | eu | Basque | 57766 |
| Swahili | sw | Niger-Congo | 22308 |
| Javanese | jv | Austronesian | 16855 |
| Estonian | et | Uralic | 46641 |
| Tagalog | tl | Austronesian | 12577 |
| Yoruba | yo | Niger-Congo | 7407 |
| Afrikaans | af | IE: Germanic | 34982 |
| Urdu | ur | IE: Indo-Aryan | 40075 |
| Burmese | my | Sino-Tibetan | 5867 |
| Georgian | ka | Kartvelian | 25455 |
| Malayalam | ml | Dravidian | 38048 |
| Marathi | mr | IE: Indo-Aryan | 31969 |
| Bulgarian | bg | IE: Slavic | 39987 |
| Kazakh | kk | Turkic | 21018 |
| Persian | fa | IE: Iranian | 74166 |

**Table 13:** The detailed results of F1 scores at the embeddings of layer-7 for mBERT and layer-13 for XLMR on BUCC dataset.

| Model | de | fr | ru | zh | Avg.(4) |
|---|---|---|---|---|---|
| mBERT | 64.3 | 62.5 | 49.5 | 53.6 | 57.5 |
| MATs-ST | 72.1 | 74.1 | 68.1 | 63.2 | **69.4** |
| MATs-DT | 72.4 | 71.8 | 63.9 | 62.0 | 67.5 |
| InteMATs | 72.7 | 70.5 | 65.9 | 65.0 | 68.5 |
| XLMR$_{large}$ | 83.8 | 73.2 | 82.9 | 67.0 | 76.7 |
| MATs-ST | 94.1 | 89.4 | 93.4 | 88.3 | **91.3** |
| MATs-DT | 90.9 | 84.4 | 89.6 | 85.7 | 87.6 |

**Table 14:** The detailed results of F1 score on RELX dataset for each languages.

| Models | Sup.(en) | de | es | fr | tr | ZS.(4) | Avg.(5) |
|---|---|---|---|---|---|---|---|
| mBERT | 61.8 | 57.5 | 57.9 | 58.3 | 55.8 | 57.4 | 58.3 |
| MATs-ST | 58.5 | 63.1 | 60.2 | 59.3 | 58.2 | **60.2** | 59.9 |
| MATs-DT | 59.6 | 58.8 | 61.9 | 57.5 | 55.8 | 58.5 | 58.7 |
| InteMATs | 63.7 | 59.9 | 61.8 | 58.1 | 57.6 | 59.4 | **60.2** |
| XLMR$_{large}$ | 63.1 | 58.0 | 59.8 | 59.5 | 59.1 | 59.1 | 59.9 |
| MATs-ST | 61.2 | 61.0 | 60.3 | 62.1 | 59.9 | 60.8 | 60.9 |
| MATs-DT | 61.3 | 62.3 | 61.5 | 59.8 | 54.7 | 59.6 | 59.9 |
| InteMATs | 61.1 | 61.8 | 62.1 | 61.5 | 61.2 | **61.7** | **61.7** |

**Table 15:** The detailed results of Acc. on PAWS-X dataset for each languages.

| Models | Sup.(en) | de | es | fr | ja | ko | zh | ZS.(6) | Avg.(7) |
|---|---|---|---|---|---|---|---|---|---|
| mBERT | 94.0 | 85.7 | 87.4 | 87.0 | 73.0 | 69.6 | 77.0 | 80.0 | 81.9 |
| MATs-ST | 94.5 | 87.1 | 87.5 | 87.3 | 74.9 | 73.2 | 79.5 | 82.9 | 83.4 |
| MATs-DT | 93.7 | 86.5 | 89.2 | 88.5 | 77.9 | 75.3 | 80.1 | 81.6 | 84.5 |
| InteMATs | 94.9 | 87.8 | 89.3 | 88.1 | 78.5 | 75.3 | 80.3 | **83.2** | **84.9** |
| XLMR$_{large}$ | 94.7 | 89.7 | 90.1 | 90.4 | 78.7 | 79.0 | 82.3 | 85.0 | 86.4 |
| MATs-ST | 95.8 | 91.8 | 92.1 | 92.4 | 80.1 | 82.3 | 84.3 | 87.2 | 88.4 |
| MATs-DT | 95.1 | 90.7 | 91.7 | 91.9 | 80.9 | 81.6 | 84.1 | 86.8 | 88.0 |
| InteMATs | 95.7 | 92.7 | 92.3 | 92.4 | 80.7 | 82.7 | 84.4 | **87.5** | **88.7** |

**Table 16:** The detailed results of Acc. on POS dataset for each languages.

| Model | Sup.(en) | af | ar | bg | de | el | es | et | eu | fa | fi | fr | he | hi | hu | id | it | ja |
|---|---|---|---|---|---|---|---|---|---|---|---|---|---|---|---|---|---|---|
| mBERT | 95.5 | 86.6 | 56.2 | 85.0 | 85.2 | 81.1 | 86.9 | 79.1 | 60.7 | 66.7 | 78.9 | 84.2 | 56.2 | 67.2 | 78.3 | 71.0 | 88.4 | 49.2 |
| MATs-ST | 95.4 | 87.0 | 53.1 | 86.9 | 87.6 | 84.7 | 87.6 | 79.8 | 63.9 | 66.2 | 79.1 | 84.8 | 53.2 | 67.4 | 78.1 | 70.8 | 88.1 | 45.4 |
| MATs-DT | 95.4 | 87.0 | 53.7 | 87.4 | 87.4 | 84.0 | 87.8 | 80.1 | 64.4 | 67.2 | 79.4 | 83.7 | 53.7 | 65.4 | 79.5 | 71.3 | 88.7 | 72.1 |
| InteMATs | 95.3 | 86.0 | 53.9 | 88.0 | 86.8 | 83.1 | 88.3 | 80.2 | 63.8 | 66.3 | 79.4 | 86.8 | 53.7 | 68.2 | 79.2 | 71.5 | 89.5 | 46.6 |
| XLMR | 96.1 | 89.8 | 67.5 | 88.1 | 88.5 | 86.3 | 88.3 | 86.5 | 72.5 | 70.6 | 85.8 | 87.2 | 68.3 | 76.4 | 82.6 | 72.4 | 89.4 | 15.9 |
| MATs-ST | 96.0 | 89.6 | 66.9 | 88.6 | 89.0 | 88.2 | 86.6 | 86.5 | 73.4 | 69.1 | 85.9 | 86.2 | 65.8 | 73.8 | 83.1 | 73.0 | 88.0 | 25.9 |
| MATs-DT | 96.1 | 89.1 | 67.7 | 88.7 | 89.1 | 87.6 | 88.9 | 86.5 | 74.6 | 70.8 | 85.9 | 87.4 | 67.2 | 74.3 | 82.6 | 72.4 | 89.3 | 37.8 |
| InteMATs | 96.1 | 89.8 | 67.0 | 89.0 | 89.4 | 87.7 | 88.3 | 86.5 | 73.9 | 69.4 | 85.8 | 87.8 | 70.3 | 76.3 | 82.5 | 72.7 | 88.9 | 27.1 |

| Model | kk | ko | mr | nl | pt | ru | ta | te | th | tl | tr | ur | vi | yo | zh | ZS.(32) | Avg.(33) |
|---|---|---|---|---|---|---|---|---|---|---|---|---|---|---|---|---|---|
| mBERT | 70.5 | 49.6 | 69.4 | 88.6 | 86.2 | 85.5 | 59.0 | 75.9 | 41.7 | 81.4 | 68.5 | 57.0 | 53.2 | 55.7 | 61.6 | 70.8 | 71.5 |
| MATs-ST | 72.5 | 49.4 | 71.7 | 88.4 | 84.8 | 86.4 | 60.4 | 77.5 | 39.4 | 83.7 | 69.8 | 55.1 | 56.0 | 55.3 | 64.2 | 71.2 | 71.9 |
| MATs-DT | 46.2 | 70.9 | 49.0 | 73.0 | 88.6 | 86.4 | 86.4 | 60.9 | 76.1 | 40.5 | 83.6 | 67.7 | 56.8 | 56.0 | 56.2 | 71.4 | 72.1 |
| InteMATs | 73.2 | 50.2 | 72.8 | 88.6 | 87.2 | 87.5 | 60.3 | 76.2 | 40.9 | 84.4 | 68.9 | 56.8 | 56.3 | 57.7 | 65.4 | **71.8** | **72.5** |
| XLMR | 78.1 | 53.9 | 80.8 | 89.5 | | 89.5 | 65.2 | 86.6 | 47.2 | 92.3 | 76.3 | 70.3 | 76.8 | 24.6 | 25.7 | 73.1 | 73.8 |
| MATs-ST | 78.9 | 52.7 | 84.4 | 89.4 | 85.7 | 90.0 | 65.0 | 86.3 | 47.1 | 94.8 | 75.6 | 68.4 | 58.2 | 34.1 | 38.3 | 74.0 | 74.7 |
| MATs-DT | 79.4 | 52.8 | 80.7 | 89.6 | 87.2 | 90.0 | 64.8 | 85.7 | 52.8 | 92.4 | 75.8 | 69 | 58.8 | 29.1 | 49 | **74.9** | 75.5 |
| InteMATs | 79.3 | 56.2 | 82.7 | 89.5 | 87.7 | 90.1 | 65.5 | 86.8 | 52.1 | 93.7 | 77.8 | 70.5 | 58.6 | 29.7 | 45.4 | **74.9** | 75.6 |

**Table 17:** The detailed results of F1/EM score on XQuAD dataset for each languages.

| Models | Sup.(en) | ar | de | el | es | hi | ru | th | tr | vi | zh | ZS.(10) | Avg.(11) |
|---|---|---|---|---|---|---|---|---|---|---|---|---|---|
| mBERT | 83.5/72.2 | 61.5/45.1 | 70.6/54.0 | 62.6/44.9 | 75.5/56.9 | 59.2/46.0 | 71.3/53.3 | 42.7/33.5 | 55.4/40.1 | 69.5/49.6 | 58.0/48.3 | 62.6/47.2 | 64.5/49.4 |
| MATs-ST | 85.4/73.8 | 62.7/45.4 | 72.7/56.6 | 63.1/45.3 | 75.5/56.6 | 57.7/42.9 | 72.9/55.4 | 44.1/33.3 | 54.0/38.8 | 70.9/51.3 | 58.4/48.6 | 63.2/47.4 | 65.2/49.8 |
| MATs-DT | 85.7/73.9 | 63.1/46.9 | 72.6/57.3 | 63.2/46.2 | 75.6/58.2 | 58.6/45.4 | 72.6/55.3 | 44.7/34.9 | 55.3/40.7 | 69.6/49.2 | 58.6/49.4 | 63.4/48.4 | 65.4/50.7 |
| InteMATs | 85.9/73.9 | 63.0/47.2 | 69.8/53.9 | 63.9/47.2 | 75.8/57.7 | 59.5/44.9 | 73.1/56.3 | 44.8/34.1 | 55.2/39.7 | 70.5/50.1 | 59.2/48.9 | **63.5/48.0** | **65.5/50.4** |
| XLMR$_{large}$ | 86.5/75.7 | 68.6/49.0 | 80.4/63.4 | 79.8/61.7 | 82.0/63.9 | 76.7/59.7 | 80.1/64.3 | 74.2/62.8 | 74.8/55.6 | 79.1/59.0 | 75.6/59.3 | | 76.6/60.8 |
| MATs-ST | 88.3/77.1 | 75.9/58.2 | 80.5/63.5 | 80.6/61.7 | 83.0/64.3 | 76.7/60.1 | 80.1/63.6 | 73.8/60.3 | 76.6/61.1 | 80.0/60.7 | 55.2/45.2 | 76.2/59.9 | 77.3/61.4 |
| MATs-DT | 87.2/75.8 | 75.6/58.2 | 80.2/63.0 | 80.4/63.4 | 82.3/64.0 | 76.9/59.8 | 80.5/64.9 | 74.7/62.4 | 76.4/60.1 | 81.5/62.7 | 56.5/48.4 | 76.5/60.7 | 77.5/62.1 |
| InteMATs | 87.4/76.4 | 75.3/58.2 | 81.7/65.3 | 80.4/61.9 | 83.1/64.9 | 76.8/59.7 | 80.1/64.5 | 74.7/61.2 | 77.1/60.8 | 80.6/61.1 | 56.6/48.4 | **76.6/60.6** | **77.6/62.0** |

**Table 18:** The detailed results of F1/EM score on TydiQA dataset for each languages.

| Models | Sup.(en) | ar | bn | fi | id | ko | ru | sw | te | ZS.(8) | Avg.(9) |
|---|---|---|---|---|---|---|---|---|---|---|---|
| mBERT | 75.3/63.6 | 62.2/42.8 | 49.3/32.7 | 59.7/45.3 | 64.8/45.8 | 58.8/50.0 | 60.0/38.8 | 57.5/37.9 | 49.6/38.4 | 57.7/41.5 | 59.7/43.9 |
| MATs-ST | 73.7/61.8 | 61.0/39.8 | 48.0/31.9 | 59.0/41.8 | 63.8/46.4 | 51.9/42.0 | 61.5/42.0 | 59.5/36.1 | 44.7/36.6 | 56.1/39.6 | 58.1/42.0 |
| MATs-DT | 74.0/63.2 | 63.4/41.2 | 55.4/40.7 | 59.3/42.1 | 63.3/47.1 | 54.6/44.6 | 60.0/40.9 | 57.1/34.1 | 49.9/38.6 | 57.8/41.2 | 59.7/43.6 |
| InteMATs | 75.6/63.1 | 64.7/43.2 | 53.5/41.8 | 59.3/42.7 | 63.6/48.7 | 53.9/44.1 | 62.3/42.5 | 58.9/36.6 | 49.8/38.4 | **58.2/42.3** | **60.2/44.6** |
| XLMR$_{large}$ | 71.5/56.8 | 67.6/40.4 | 64.0/47.8 | 70.5/53.2 | 77.4/61.9 | 31.9/10.9 | 67.0/42.1 | 66.1/48.1 | 70.1/43.6 | 64.3/43.5 | 65.1/45.0 |
| MATs-ST | 73.1/61.1 | 73.3/52.9 | 64.2/45.1 | 72.5/54.7 | 78.5/62.3 | 59.7/45.7 | 66.7/39.2 | 74.6/55.6 | | 70.1/51.8 | 70.1/51.8 |
| MATs-DT | 73.9/61.1 | 74.6/55.4 | 66.4/52.1 | 73.9/57.5 | 78.9/63.5 | 59.7/46.4 | 67.1/38.8 | 69.2/50.9 | 74.8/53.7 | 70.9/53.3 | 70.9/53.3 |
| InteMATs | 73.8/61.3 | 75.7/55.5 | 66.8/53.6 | 74.3/58.4 | 78.8/64.6 | 59.2/46.3 | 67.5/38.1 | 68.5/50.6 | 75.5/54.6 | **70.8/52.7** | **71.1/53.7** |