# OpenReview forum: "InteMATs: Integrating Granularity-Specific Multilingual Adapters for Cross-Lingual Transfer"
_EMNLP/2023/Conference — EMNLP 2023 Findings_

### Official Review · Reviewer_27bY · 2023-07-26

**Typos Grammar Style And Presentation Improvements:** Table 4, ZS(32)
**Soundness:** 3

**Excitement:**

3: Ambivalent: It has merits (e.g., it reports state-of-the-art results, the idea is nice), but there are key weaknesses (e.g., it describes incremental work), and it can significantly benefit from another round of revision. However, I won't object to accepting it if my co-reviewers champion it.

**Paper Topic And Main Contributions:**

The main goal of this work is to improve cross-lingual transfer in the pretrained multilingual models. To achieve this goal, this work proposes to integrate multilingual adapters of trained on the tasks of different granularities: sentence level + document level adapters. These adapters are then combined via AdapterFusion mechanism. Moreover, inspired by previous work (form CV community) authors suggest that contrastive learning favours global features learning (as opposed to Masked LM training that rather favours local features). Therefore both sentence-level and document-level adapters are trained with Contrastive learning. For training such adapters authors create entity-aligned  dataset  gathered from Wikipedia covering 43 languages in total. The dataset created at sentence-level (first sentence describing each entity)  and document-level (full document).

Thus 3 main contributions of this paper can be summarized as
1) creating of multilingual entity-aligned dataset for sentence-level and document-level granularities
2) Fusion of sentence-level adapters and document-level adapters in a single module
3) Contrastive learning of adapters (as opposed to MLM training usually used)

Overall impression: I find that  the ideas this work is advancing are interesting, the experimental results and some of the ablation studies are quite interesting as well.  But I am left with the feeling that the experimental results do not connect well to the originally stated claims.  It is maybe just a matter of the 1-2 more experiments, or better experiments explanation, but in the current status it is quite hard to make this connection.

**Questions For The Authors:**

My questions are those raised in reasons to reject:  I would like to better understand impact of each of the experimental choice made by authors

1) training on entity aligned dataset should definitely improve cross-lingual transfer, would it be the main factor of the gains you observe on low-resource languages?

2) I really liked the arguments and idea for contrastive loss training, but I can't see any experimental proof supporting these arguments


3) You seem to use average between the first and the last layers as a sample representations (not sure if it is only true for STS task or other tasks as well?). However, the plot on FIgure 3 seems to suggest that the "best" repreresentations for cross-lingual transfer seem to be somewhere in the middle.  Quite similar observations were made also at https://arxiv.org/abs/2212.09535.
 How the results would change if you used representations from the layers 6-9 (xlmr-base)/12-16 (xlmr-large) instead?

**Reasons To Accept:**

- all the three proposed contributions (datasets, adapters of different granularities, and contrastive learning) are interesting and useful contributions, the choices are also well motivated and make total sense to try them
- a lot of baselines were considered and many tasks were addressed, an overview of different methods in the same table is interesting in itself (better presenting these different baseline would help the readability however for the readers who are not necessarily aware of the differences between them)
 - Authors provide an interesting analysis showing the relevance of document-level adapters for the tasks (XSquad, TydiQA)

**Reasons To Reject:**

- this work compares proposed method to many other models previously proposed in the litterature. This comparison is quite exhaustive in my point of view and appreciate authors consideres so many baseline
- However, the major drawback for me is that the contribution of each of the component proposed in this work in the overall final result is not clear: does contrastive learning really favour global features in case of NLP tasks? Or all the benefits we see are from document+sentence level adapters fusion? What if we just train classical adapters on the whole entity-aligned corpus (without dsitinguishing the granulrity): what would be the quality of cross-lingual transfer? If we train such adapters with contrastive loss would it improve cross-lingual transfer? If we distinguish between sentence-level and document-level adapters would it help further? In the current experimental settings authors compare with many different previously proposed methods, but I would assume those baselines were trained on different datasets, and with different training objectives, therefore it is hard to draw meaningful conclusions from these results.

**Reproducibility:**

3: Could reproduce the results with some difficulty. The settings of parameters are underspecified or subjectively determined; the training/evaluation data are not widely available.

**Reviewer Confidence:**

3: Pretty sure, but there's a chance I missed something. Although I have a good feel for this area in general, I did not carefully check the paper's details, e.g., the math, experimental design, or novelty.

---

> ### Author Rebuttal · Authors · 2023-08-29
>
> We thank the reviewer for recognizing the value and contributions of our paper to the field. We will revise the experiments' explanations accordingly.
>
> - For the contribution of each of the components in the overall final result, we address specific concerns below:
>
> Q1. **[Does contrastive learning (CL) really capture global features?]**: While this question is beyond the scope of this paper, we’d like to provide our thoughts as follows.
>
> Based on existing research, a quick answer is Yes. The recent findings (Park et al., 2023) from the CV community provided visualized comparisons between the features captured by CL and the features captured by MIM. In NLP, however, it’s not straightforward to define and visualize global and local features from texts. In view of previous results, e.g., SimSCE shows that using CL to continue the pretraining of an MLM-trained language model can produce sentence representations that align better in a geometric space, as evaluated on similarity-based tasks such as sentence retrieval; mSimCSE and InfoXLM extend the use of CL to multilingual settings and show that it is similarly useful to enhance cross-lingual alignment of XLMR; MLKG demonstrated that CL can also be used to teach adapters to learn knowledge graph triples that inherently have certain geometric properties. Therefore, this paper extends from previous findings and uses CL as a pretraining objective to ensure a better cross-lingual alignment capability for our InteMATs. Figure 4 in the Appendix compares the similarity between aligned embeddings, demonstrating that such capability has been effectively acquired by InteMATs.
>
> Theoretically, the loss function of CL uses distance-based measures and it enforces the model to make relevant data closer and irrelevant data far away from each other. This requires the sentence or document representations to emphasize **a set of features** that can make such distinctions in general. Local features, however, only focus on a very small part of the input. It is impossible for a certain kind of local features to clearly distinguish one language from another.
>
> Q2. **[Are all the benefits we see resulting from fusing these adapters?]**:
>
> Table 8 presents the examinations on each of the granularity-specific adapters: MATs-ST and MATs-DT. In general, both sentence-level and document-level adapters can improve the performance of an MLM-trained multilingual model, i.e., mBERT and XLMR. MATs-ST excels at sentence-level tasks (e.g., BUCC and RELX) while MATs-DT excels at document-level tasks (e.g., XQUAD and TydiQA). Fusing these adapters confers the most benefits and the reasons behind this is exhaustively studied in Section 5.10.1 and 5.10.2.
>
> We show that InteMATs does effectively leverage the corresponding granularity-specific adapters for any given task (Figure 2) without knowing the text granularity in advance. The benefits that InteMATs has brought to a given MLLM increase from the first layer to the last layer (Figure 3), indicating that such capability is mainly acquired on the last layers.
>
> Q3. **[How about training classical adapters (using MLM) on the whole entity-aligned corpus (without distinguishing the granularity)?]**:
>
> The whole corpus refers to the document-level corpus. We have a prior work that focuses on exploring a more diverse range of text granularities which has just been accepted. Our prior work extensively compared all kinds of pretraining objectives and we found that CL works the best. We bring the results here: training classical adapters on document-level corpus **using CL produces 3-6% higher performance than using MLM** on TydiQA, BUCC, and WK3l. We will revise our explanations and make reference to this paper in our final version.
>
> We want to emphasize that this paper aims to contribute an approach for **automatically integrating** different kinds of granularity-specific multilingual adapters for **any given task**, and examine whether such integration chooses the appropriate adapters that comply with our intuition.
>
> Q4. **[switch MLM to CL for training adapters]**:
>
> Refer to point 3. The result of MATs-DT in Table 8 is obtained by training adapters with CL on the whole corpus.
>
> Q5. **[Then, consider sentence-level and document-level adapters]**:
>
> Please refer to Table 8 for the results of incorporating either sentence-level (MATs-ST), document-level adapters (MATs-DT), or integrating both (InteMATs).
>
>
> - We further address the additional questions below:
>
> Q1. **[Is the pretraining corpus the main factor of the gains on low-resource languages?]**:
>
> No. Both pretraining and integration are necessary. Our prior work shows that pretrained adapters outperform randomly initialized adapters by up to 38% on Tatoeba when conditioned on XLMR. (Please refer to the Q2 of Reviewer 2.) This paper shows that different tasks favor different kinds of granularity-specific multilingual adapters and an effective integration can achieve consistent improvements. Tables 13, 14, 15, and 16 in the appendix present the results on very individual language.
>
> Q2. **[Provide experimental proof supporting the idea for contrastive loss training]**:
>
> In the appendix, Figure 4 visually presents how CL enhances the cross-lingual alignment for MLLMs.	As we mentioned earlier, we have an extensive experimental comparison between CL and MLM in our prior work and we will make reference accordingly in the final version. Again, the proof of CL is not the primary goal of this paper.
>
> Q3. **[Why use the average between the first and the last layers as a sample representation for the STS task?]**:
>
> We follow mSimSCE to use the average of the first and last layer representations **only** for the STS task for **fair comparison**. In the mSimCSE paper, authors claim that such practice produces the best performance on STS, and the reasons behind were not provided.
>
> **[How about using representations from the layers 6-9 (xlmr-base)/12-16 (xlmr-large) instead?]**:
> We appreciate this constructive suggestion and we’d like to explore it in future work. This paper is not limited to the evaluations on the Semantic Textual Similarity task only, we have other 4 kinds of cross-lingual tasks: sentence retrieval, sentence classification, syntactic analysis, and QA. For the rest tasks, we just use the last layer representations.
>
> However, these deep questions, while worth discussing, do not necessarily override the validity or undermine the soundness of our approach. We thank again the reviewer for driving our attention to so many details and we will make clarifications accordingly in the final version.

---

### Official Review · Reviewer_JJrR · 2023-08-03

**Soundness:** 3

**Excitement:**

3: Ambivalent: It has merits (e.g., it reports state-of-the-art results, the idea is nice), but there are key weaknesses (e.g., it describes incremental work), and it can significantly benefit from another round of revision. However, I won't object to accepting it if my co-reviewers champion it.

**Paper Topic And Main Contributions:**

This paper presents a multilingual encoder and aims to improve accuracies of NLP tasks in low-resource languages and long contexts.

The characteristics of the proposed method is as follows.
1) An adapter for sentences, an adapter for documents, and a fusion module of both adapters were integrated into large pretrained models.
2) For the corpora, the authors used sentences including entity alignment information and documents obtained from Wikipedia articles in 43 languages.
3) Only the adapters and fusion module were tuned by contrastive learning that contrast languages.

In their experiments, accuracies of various tasks (sentence-level and document-level) were improved.

**Questions For The Authors:**

A) Sec. 4.1:
A characteristic of the proposed method is using a corpus that include "entity alignment information."  I could not imagine the format of the corpus.  Please show examples to readers.  I assume that the entity alignment information means Wikipedia titles among multiple languages or link information in sentences. The example in Fig. 1 (b) seems to be no link information.

**Reasons To Accept:**

* Accuracies of various tasks were improved.
* In the document-level tasks, the accuracy was improved because two adapters effectively worked.

**Reasons To Reject:**

* Although the accuracies of various tasks were improved, they were system comparison.  It is unclear what components of the proposed method (the corpus, the adapters and fusion module, and contrastive learning) worked to improve accuracy.  More analyses are necessary.

* Similar to the previous comments, The long context problem was alleviated by the proposed model in Table 8 and Figure 2.  However, in the low-resource problem, it is unclear which component was effectively worked to improve the accuracies in Tables 3-7, the corpus or contrastive learning.
Additional experiments are necessary to clarify effectiveness of the components.  For example, the base pretrained models (mBERT and XLM-R) are additionally learned using the authors' corpora, and then fine-tune the models to the tasks.

**Reproducibility:**

3: Could reproduce the results with some difficulty. The settings of parameters are underspecified or subjectively determined; the training/evaluation data are not widely available.

**Reviewer Confidence:**

3: Pretty sure, but there's a chance I missed something. Although I have a good feel for this area in general, I did not carefully check the paper's details, e.g., the math, experimental design, or novelty.

**Typos Grammar Style And Presentation Improvements:**

* The size of letters in the tables and figures are too small.
This is not reader-friendly.

---

> ### Author Rebuttal · Authors · 2023-08-29
>
> We thank the reviewer for the comments and address each of them below:
>
> Q1. **[Which of the corpus, the adapters, fusion module, and contrastive learning worked to improve accuracy?]**:
>
> We want to clarify that comparing every alternative choice for every component is not the primary goal of this paper. The value of this paper is providing an approach for **automatically integrating** granularity-specific adapters for **any given** task without knowing the text in advance. Examining every step from the beginning produces **too many results** to fit the allowed page limit, and we have put these extensive results in a prior work that has just been accepted. We will put the reference in the final version.
>
> - **[The role of pretraining corpus]**: It is widely accepted that pretraining generally provides a better-initialized model than a randomly initialized one. We additionally report some results here: Equipping mBERT with randomly initialized adapters produces 2% lower performance on Tatoeba, 3% lower on RELX, and 5% lower on TydiQA than with adapters pre-trained on our corpus. Moreover, randomly initialized adapters even undermine the performance of mBERT on many benchmarks. We will release the corpus for future studies.
>
>   | Model | PAWS-X | Tatoeba | RELX | TydiQA |
>   | ---- | ---- |---- |---- |---- |
>   | mBERT | 81.9 | 34.6 | 58.3 | 59.7/43.0 |
>   | +Initilized | 83.6 |34.1 | 57.9 | 55.8/43.7 |
>   | InteMATs | 84.9 | 36.0 | 60.2 | 60.2/44.6 |
>
> - **[The adapters and fusion module]**: We believe we have proved this in Table 8. More analysis can be found in Figures 2, 3, and 4.
>
> - **[The role of contrastive learning (CL)]**: The advantage of using CL as a pretraining objective has been widely studied in previous research. SimSCE is a CL-trained model that continues the pretraining of an MLM-trained language model on additional corpora. mSimSCE and InforXLM are CL-trained multilingual models. MLKG employs CL to incorporate knowledge graphs into MLLMs. etc. The exact result of comparing the choice of CL and MLM for MATs-ST or MATs-DT can be found in our prior work where we demonstrate that CL continues to excel in this problem.
>
> Q2. **[In the low-resource problem, is the corpus or contrastive learning effectively worked to improve the accuracies in Tables 3-7 ?]**:
>
> We extensively examined the effect of the pretraining corpus and the CL pretraining objective in our prior work. We will make references to relevant conclusions accordingly. However, this paper aims to present a novel InteMATs approach for automatically integrating different granularity-specific adapters. We have compared InteMATs against classical and most recent baselines on five different downstream tasks. The advantages of InteMATs against domain gap (Table 6) and against  **low-resource languages** (Table 7) have already been discussed. The effectiveness of **integration** has also been demonstrated in Table 8, Figures 2, 3, and 4.
>
> - **Does the pretraining corpus benefit low-resource languages?** Yes. We additionally report the results of pre-trained and randomly initialized adapters on Tatoeba as follows. We found that the randomly initialized adapters not only perform worse than pre-trained ones (-38% on XLMR) but can also undermine the performance of MLLMs (-2% on mBERT), indicating that the parallel data can teach adapters with cross-lingual alignment knowledge, particularly useful for XLMR. More results on other low-resource languages can be found in the Appendix from Table 12-16.
>
>   | Model | Avg. | Model | Avg. |
>   | ---- | ---- | ---- | ---- |
>   | mBERT | 24.1 | XLMR | 24.5 |
>   |+random | 21.9 | +random | 24.3 |
>   | InteMATs | 24.2 | InteMATs | 62.3 |
>
> Q3. **[How about training the base pretrained models using the authors' corpora, and then fine-tune the models to the tasks?]**:
>
> Again, we want to clarify that the corpus is not the only contribution that we make in this paper. We have this result in our prior work where we have found that finetuning the entire mBERT model on the whole pretraining corpus produces 6% lower performance than MATs-ST on BUCC, 2% lower than MATs-ST on RELX, and 1% lower than MATs-DT on XQUAD. Since both MATs-ST and MATs-DT underperform InteMATs in this paper, finetuning the entire model on our corpus must work worse than InteMATs.
>
> Q4. **[About the example of entity-aligned information]**:
>
> We have given an example of sentence-level corpus in Figure 1 (b). The **ChatGPT** item refers to an aligned entity. More details have been described in Section 5.1.

---

### Official Review · Reviewer_R9P3 · 2023-08-07

**Soundness:** 3

**Excitement:**

2: Mediocre: This paper makes marginal contributions (vs non-contemporaneous work), so I would rather not see it in the conference.

**Missing References:**

1. Ansell et al., MAD-G: Multilingual Adapter Generation for Efficient Cross-Lingual Transfer, findings of EMNLP 2021
2. Wang et al., Efficient Test Time Adapter Ensembling for Low-resource Language Varieties, findings of EMNLP 2021
3. Baziotis et al., Multilingual Machine Translation with Hyper-Adapters, EMNLP 2022


On the cross-lingual transferability of monolingual representations is cited twice.

**Paper Topic And Main Contributions:**

The paper proposed InterMATs, a method that integrates multilingual adapters trained at different granularity levels, to address the low performance of multilingual language models (MLLMs) on zero-shot low-resource languages, particularly when dealing with longer contexts.

The approach involves pre-training sentence-level and document-level adapters using the contrastive learning framework on a multilingual parallel dataset. Sentence-level multilingual adapters (MATs-ST) process texts at the sentence level, while document-level multilingual adapters (MATs-DT) capture global information from document-level texts. The adapters are then integrated using AdapterFusion, enhancing the representations of a fixed MLLM.

Extensive experiments are conducted to evaluate the effectiveness of InteMATs, and the results show performance enhancements across various cross-lingual understanding tasks while updating fewer parameters than other MLLMs.

**Reasons To Accept:**

1. Sound Approach: InteMATs integrates multilingual adapters trained at different granularity levels (sentence-level and document-level). This approach has the potential to address the limitations of multilingual language models (MLLMs) on zero-shot low-resource languages and longer contexts, offering a unique and promising solution to a significant problem in the field.

2. Extensive experiments: The paper presents extensive experiments evaluating the effectiveness of InteMATs across various cross-lingual understanding tasks. The reported performance enhancements compared to existing MLLMs demonstrate the method's efficacy. Moreover, the reduction in the number of parameters updated during the integration process highlights the efficiency gains achieved by InteMATs, making it a valuable contribution to the field.

**Reasons To Reject:**

1.Limited Comparative Evaluation: The use of paired data during InteMATs pretraining results in relatively modest improvements on each dataset. The comparison with unsupervised methods like mBERT, XLMR, and mSimCSE is provided as baseline; however, a fairer comparison could be achieved by applying contrastive learning to XLMR first. The absence of a comprehensive benchmarking analysis and failure to address potential trade-offs in model accuracy or other aspects of performance undermines the significance of the reported results.

2. Missing Citations: Despite numerous related papers on multilingual adapters, the authors overlook citing them in their work. Some of the missing citations are listed below for reference. (Please refer to Missing References)



**Reproducibility:**

4: Could mostly reproduce the results, but there may be some variation because of sample variance or minor variations in their interpretation of the protocol or method.

**Reviewer Confidence:**

4: Quite sure. I tried to check the important points carefully. It's unlikely, though conceivable, that I missed something that should affect my ratings.

**Typos Grammar Style And Presentation Improvements:**

To enhance the clarity and understanding for readers, I recommend including a brief introduction to each baseline method used in the experiments. This introduction should outline the key features and training methodologies of each baseline and highlight any significant differences between the baselines and InteMATs. Additionally, it is crucial to cite the original papers for each baseline to allow readers to access further details and gain a deeper understanding of the comparison. By addressing these aspects, the paper can offer a more comprehensive and well-informed evaluation of InteMATs against the selected baselines.

---

> ### Author Rebuttal · Authors · 2023-08-29
>
> We thank the reviewer for constructive suggestions and positive comments on the soundness and significance of our approach.
>
> Q1. **[A fairer comparison: applying contrastive learning (CL) to XLMR]**:
>
> - **[Full-model finetuning of MLLMs with CL]**: The baseline models, mSimCSE and InfoXLM, were obtained by applying CL to train the entire XLMR model on multilingual data pairs. They have already proved the effectiveness of using CL as a pretraining objective to enhance the cross-lingual representations of MLLMs. Nevertheless, they need to train the entire MLLM backbone and therefore consume much more language data. E.g., InfoXLM uses 93 languages while ours uses only 43 languages. The cross-lingual performance of our parameter-efficient solution, InteMATs, is 4-7% higher than mSimSCE and 7-20% higher than InfoXLM on BUCC and Tatoeba.
>
> - **[Parameter-efficient tuning of MLLMs with CL]**: The baseline, MLKG, has proved the effectiveness of using CL as a pretraining objective to incorporate knowledge into MLLMs with adapters trained on knowledge graph triples. Yet, the pretraining corpus of MLKG is 36G in size. Our approach employs only 1.95G pretraining data but achieved around 1 percent higher than MLKG on zero-shot languages of the RELX and XQUAD benchmarks. Please refer to Table 10 for a detailed comparison of parameters and data consumption.
>
> - **[Our value]**: We put forth the hypothesis that text granularities may play a crucial role in determining the effectiveness of cross-lingual alignment during pre-training for different tasks. Figure 2 shows that sentence-level tasks put higher weights on sentence-level adapters while document-level tasks put higher weights on document-level adapters, which supports the importance of differentiating text granularities in pretraining for different tasks. To avoid manual differentiation, we propose to automatically learn the weights to adapt to different tasks regardless of text granularities.
>
> Q2. **[Missing references]**:
>
> - Thanks for referring us to these references. We will include them in the section of Related Works. In short, Ansell et al. and Wang et al. proposed different approaches to train **language-specific** adapters while we contribute a new kind of generalized **language-agnostic** adapters for MLLMs. Language-specific adapters require training a different set of adapters for every different language while ours do not need. We show that ours can even improve a bit (averagely 15-38% for XLMR) on unseen languages (Section 5.9). All three references focus on limited evaluations such as syntactic analysis and machine translation (Baziotis et al.) while we conducted a more comprehensive evaluation on five kinds of cross-lingual tasks: syntactic analysis, textual similarity, sentence retrieval, sentence classification, and question-answering.
>
> Q3. **[Introduction of baselines]**:
>
> - Thanks for your suggestion. Due to the page limit, we only managed to fit a summary of the baselines in the current version. We will revise Section 5.2 to provide more details and comparisons in the final version.

---

### Meta-Review · Area_Chair_Fdau · 2023-09-18

**Recommendation:** 3

**Metareview:**

The paper introduces, a novel approach to improving cross-lingual transfer in pretrained multilingual models. The method leverages multilingual adapters trained on different granularities (sentence-level and document-level) and combines them using AdapterFusion. These adapters are pretrained using contrastive learning on a newly created multilingual entity-aligned dataset. The paper demonstrates the effectiveness of their approach through extensive experiments, achieving state-of-the-art results on various cross-lingual tasks.

Reviewers raised several concerns, including the need for clearer delineation of contributions, a desire to decouple the impacts of document-level adapters and contrastive learning, and a request for more detailed explanations of experimental choices, particularly regarding low-resource languages.
The authors responded to the concerns by providing clearer explanations of their contributions, emphasizing the non-trivial nature of fusion, and explaining the challenges of decoupling factors tied to the pretraining objective. They addressed the request for more clarity on the impact of experimental choices and offered additional data regarding low-resource languages.

Overall, while the reviewers had concerns about the clarity of contributions and the impact of individual factors, the authors' responses provided reasonable justifications and clarifications.

---

### Decision · Program_Chairs · 2023-10-07

**Decision:**

Accept-Findings

**Comment:**

The paper introduces, a novel approach to improving cross-lingual transfer in pretrained multilingual models. The method leverages multilingual adapters trained on different granularities (sentence-level and document-level) and combines them using AdapterFusion. These adapters are pretrained using contrastive learning on a newly created multilingual entity-aligned dataset. The paper demonstrates the effectiveness of their approach through extensive experiments, achieving state-of-the-art results on various cross-lingual tasks.

Reviewers raised several concerns, including the need for clearer delineation of contributions, a desire to decouple the impacts of document-level adapters and contrastive learning, and a request for more detailed explanations of experimental choices, particularly regarding low-resource languages.
The authors responded to the concerns by providing clearer explanations of their contributions, emphasizing the non-trivial nature of fusion, and explaining the challenges of decoupling factors tied to the pretraining objective. They addressed the request for more clarity on the impact of experimental choices and offered additional data regarding low-resource languages.

Overall, while the reviewers had concerns about the clarity of contributions and the impact of individual factors, the authors' responses provided reasonable justifications and clarifications.